# Identification of a *Musashi2* translocation as a novel oncogene in myeloid leukemia

Kyle Spinler[1,2], Michael Hamilton[1,2], Jeevisha Bajaj[1,2], Yutaka Shima[1,2], Emily Diaz[1,2], Marcie Kritzik[1,2,3,4], Tannishtha Reya[1,2,3,4]*

[1]Departments of Pharmacology and Medicine, University of California San Diego School of Medicine La Jolla, San Diego, United States; [2]Moores Cancer Center, University of California San Diego School of Medicine, San Diego, United States; [3]Herbert Irving Comprehensive Cancer Center, Columbia University Medical Center, New York, United States; [4]Department of Physiology and Cellular Biophysics, Columbia University Medical Center, New York, United States

## eLife assessment

The study presents **important** findings on the role of MSI2-HOXA9 translocation in chronic myeloid leukemia. The authors provide **convincing** evidence supporting the role of this translocation in leukemogenesis by using elegant mouse modeling and in vitro mechanistic studies. Consistent with the reviews, the studies can be strengthened with further murine and cell line experiments. Raw sequencing data for this manuscript are not available as the authors could not locate the files after moving institutions. Expression data for the RNA Seq data are available.

*For correspondence:
tr2726@cumc.columbia.edu

**Competing interest:** The authors declare that no competing interests exist.

**Abstract** Myeloid leukemias, diseases marked by aggressiveness and poor outcomes, are frequently triggered by oncogenic translocations. In the case of chronic myelogenous leukemia (CML), the BCR-ABL fusion initiates chronic phase disease with second hits allowing progression to blast crisis. Although Gleevec has been transformative for CML, blast crisis CML remains relatively drug resistant. Here, we show that *MSI2-HOXA9*, a translocation with an unknown role in cancer, can serve as a second hit in driving bcCML. Compared to BCR-ABL, BCR-ABL/MSI2-HOXA9 led to a more aggressive disease in vivo with decreased latency, increased lethality, and a differentiation blockade that is a hallmark of blast crisis. Domain mapping revealed that the *MSI2* RNA binding domain RRM1 had a preferential impact on growth and lethality of bcCML relative to RRM2 or the HOXA9 domain. Mechanistically, MSI2-HOXA9 triggered global downstream changes with a preferential upregulation of mitochondrial components. Consistent with this, BCR-ABL/MSI2-HOXA9 cells exhibited a significant increase in mitochondrial respiration. These data suggest that MSI2-HOXA9 acts, at least in part, by increasing expression of the mitochondrial polymerase POLRMT and augmenting mitochondrial function and basal respiration in blast crisis. Collectively, our findings demonstrate for the first time that translocations involving the stem and developmental signal MSI2 can be oncogenic and suggest that MSI, which we found to be a frequent partner for an array of translocations, could also be a driver mutation across solid cancers.

## Introduction

Myeloid leukemias are a heterogeneous group of cancers, many of which have limited treatment options and are associated with poor prognosis. Chronic myeloid leukemia (CML) is characterized by the accumulation of myeloid precursors and mature myeloid cells and is driven by translocations between the *BCR* serine/threonine kinase gene and the *ABL* tyrosine kinase gene, resulting in a

constitutively active ABL tyrosine kinase. Although imatinib mesylate effectively blocks the activity of the BCR-ABL kinase and has been used to treat CML, it is not curative because the cancer stem cells that propagate CML are no longer addicted to kinase signaling and are thus resistant to therapy (*Shah et al., 2002*). Progression from CML to blast crisis CML (bcCML) occurs with acquisition of an additional oncogenic hit; this event is associated with a rapid expansion of immature blast cells and a concomitant increase in imatinib resistance such that only 50% of patients are responsive to therapy. The specific secondary translocations and mutations that have been reported in bcCML include activation of oncogenes such as *RAS* and *MYC* (*Deininger et al., 2000*), translocations such as *NUP98-HOXA9*, and mutations in tumor suppressors such as RB1, TP53, and CDKN2A (*Ahuja et al., 2001*; *Johansson et al., 2002*; *Mayotte et al., 2002*; *Yamamoto et al., 2000*). Although multiple mutations are seen in bcCML, the only mouse model of bcCML that exists was created with a combination of *BCR-ABL* and the *NUP98/HOXA9* gene fusion (associated with t(7;11)(p15;p15); *Dash et al., 2002*; *Kawakami et al., 2002*).

A new translocation between the cell fate determinant Musashi2 (*MSI2*) and *HOXA9* was reported several years ago as a novel genetic lesion present in bcCML patients (*Barbouti et al., 2003*). In this study, 33 CML-accelerated phase/blast crisis patient samples were used for a multicolor FISH study. Among this group, two cases harbored a translocation, t(7;17)(p15;q23), resulting in a *MSI2/HOXA9* fusion gene. This in-frame fusion transcript retained both MSI2 RNA recognition domains and the HOXA9 homeobox domain, raising the potential for previously uncharacterized roles in CML progression. This translocation was of particular interest because it was the first reported mutation involving MSI. The MSI family of RNA binding proteins is comprised of two members, MSI1 and MSI2, each containing two RNA recognition motifs (RRMs) separated by a short linker region in the N-terminal half of the protein (*Sakakibara et al., 1996*). Functionally, MSI can either suppress translation by sequestering poly-A binding protein (PABP) or activate translation by stabilizing the RNA and recruiting the poly (A) polymerase GLD2 (*Johansson et al., 2002*) after binding to consensus sequences in the 3′ untranslated region (3′ UTR) of mRNA. Together, these activities result in modulation of a diverse set of genes that regulate stem and progenitor cell growth in multiple tissues, including *Numb*, an antagonist of Notch signaling (*Imai et al., 2001*). Msi2 overexpression led to elevation of the self-renewal genes *Hoxa9* and *Hoxa10*, the Sonic hedgehog and Notch signaling components *Gli1* and *Hes1*, and *Cyclin D1* or *Ccdn1* and *Cdk2* (*Hope et al., 2010*). In addition to its role in normal development (*Hope et al., 2010*; *Nakamura et al., 1994*; *Sakakibara et al., 2002*), MSI proteins have emerged as critical mediators and dependencies of both solid cancers and hematologic malignancies. In pancreatic cancer, loss of MSI2 has been shown to lead to a defect in progression from PanIN to adenocarcinoma (*Fox et al., 2016*), exerting its impact through powerful oncogenes such as *Met* and epigenetic regulators such as BRD4 and HMGB2. In colon adenomas driven by APC loss-of-function mutations, MSI2 acts as an inhibitor of known tumor suppressors, including LRIG1, BMPR1A, CDKN1A, and PTEN and sustained activation of mTORC1 (*Wang et al., 2015*). In the context of hematologic malignancies, MSI2 is required for the development and progression of myeloid leukemias (*Ito et al., 2010*; *Kwon et al., 2015*). The loss of MSI2 reduced cancer stem cell frequency, increased differentiation, and impaired propagation of bcCML in mouse models in vitro and in vivo. Further, overexpression of an *MSI2* transgene in bone marrow cells expressing BCR-ABL led to increased tumor burden (*Kharas et al., 2010*). MSI2 is also highly upregulated during human CML progression, and its expression serves as an early indicator of poor prognosis not only in leukemias but also in solid cancers like lung cancer (*Makhov et al., 2021*). However, while these studies have collectively identified MSI genes as clear dependencies in cancers and have highlighted their potential utility as a prognostic, whether naturally occurring MSI translocations and mutations can serve as oncogenes in driving cancer growth remains unknown.

Here, we have used the *MSI2-HOXA9* fusion gene to show that this translocation, found in patients with bcCML, can cooperate with BCR-ABL to drive progression to bcCML, demonstrating for the first time that the MSI gene alterations that occur in patients can serve as oncogenes. Further, the MSI2 RRMs were essential to the oncogenicity of this translocation, and they act to control bcCML by enhancing mitochondrial energetics. Deletion of the RRM1 abrogated the enhanced cell growth conferred by the MSI2-HOXA9 fusion in vitro and in vivo. Additionally, the MSI2-HOXA9 fusion resulted in cells that showed a rise in maximum mitochondrial respiration rate likely due to the concomitant increase in the mitochondrial polymerase, POLRMT, whose elevated expression may be driven by

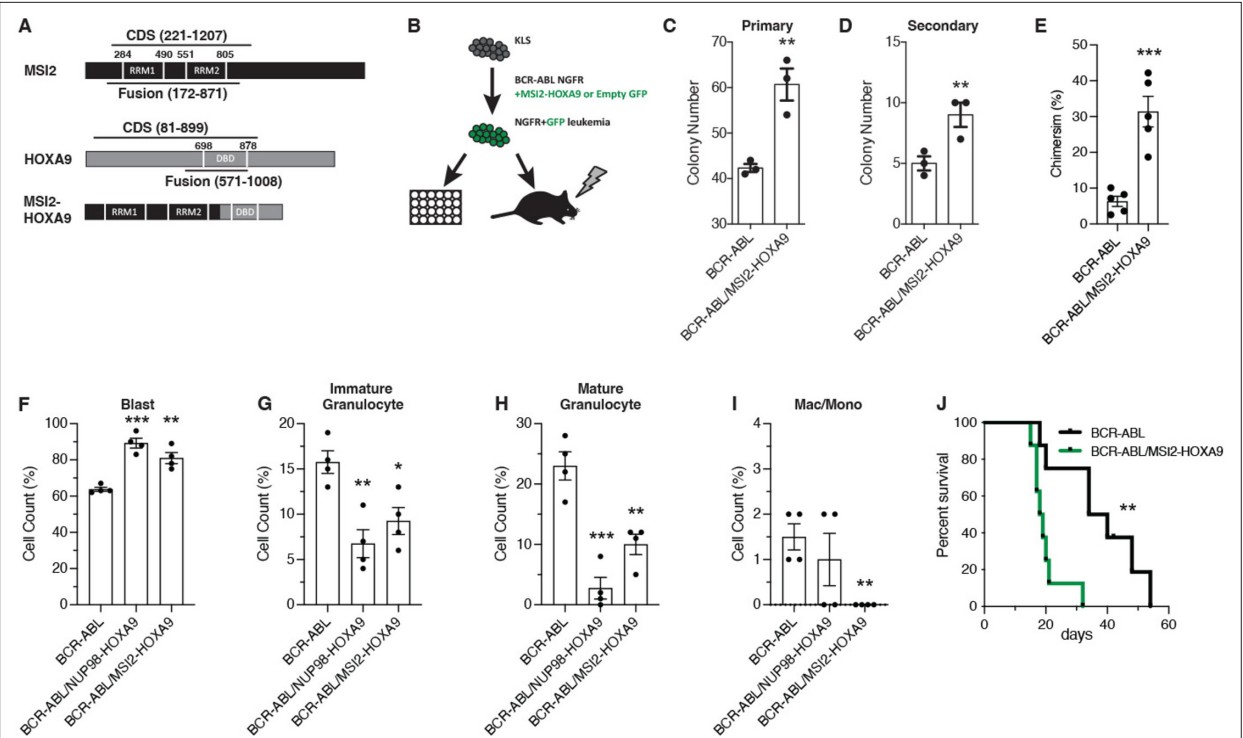

**Figure 1.** MSI2-HOXA9 expression leads to cancer cell growth advantage and differentiation arrest. (**A**) Schematic of the MSI2-HOXA9 fusion protein. The fusion retains both MSI2 RNA binding domains and the HOXA9 DNA binding domain. (**B**) Experimental design for in vitro colony assay or in vivo transplantation of BCR-ABL/Control or BCR-ABL/MSI2-HOXA9 expressing KLS cells. Transplanted mice were subsequently assessed for chimerism, survival, and cancer cell differentiation. (**C**) Primary colony assay of BCR-ABL/Control or BCR-ABL/MSI2-HOXA9 expressing KLS cells. **p=0.007 (n=3 technical replicates). (**D**) Secondary colony assay of BCR-ABL/Control or BCR-ABL/MSI2-HOXA9 expressing KLS cells. **p=0.009 (n=3 technical replicates). (**E**) Chimerism of BCR-ABL/Control or BCR-ABL/MSI2-HOXA9 transplanted cells at 13 days. ***p=0.0005 (n=5 mice per group). (**F–I**) Lin- BCR-ABL/Control, BCR-ABL/NUP98-HOXA9, or BCR-ABL/MSI2-HOXA9 expressing cells were sorted from primary transplants and used to generate cytospins that were then stained with Giemsa and May-Grunwald stains (n=4 mice per group). (**F**) Quantification of blast cells. **p=0.001, ***p=0.0001. (**G**) Quantification of immature granulocytes. *p=0.01, **p=0.004. (**H**) Quantification of mature granulocytes. **p=0.004, ***p=0.0005. (**I**) Quantification of differentiated macrophages and monocytes. **p=0.002. (**J**) Survival of mice transplanted with BCR-ABL/Control (n=6 mice) or BCR-ABL/MSI2-HOXA9 (n=7 mice) expressing KLS cells. **p=0.002. Two-tailed unpaired Student's t-tests were used to determine statistical significance.

The online version of this article includes the following figure supplement(s) for figure 1:

**Figure supplement 1.** Analysis of BCR-ABL/MSI2-HOXA9 leukemia.

**Figure supplement 2.** Examples of flow cytometry gating strategies.

the abnormal localization of MSI2-HOXA9 to the cell nucleus. Collectively, these data demonstrate a novel *MSI2* translocation-driven downstream event that leads to a more energetic, differentiation-arrested aggressive state, thus triggering CML progression to bcCML.

## Results

### MSI2-HOXA9 confers a growth advantage and arrests differentiation

To test the oncogenic activity of the MSI2-HOXA9 fusion, we generated an *MSI2-HOXA9* gene reflecting the breakpoint reported in CML patients (t(7:17)(p15;q23)). Specifically, we fused the amino terminus of MSI2 (containing both RNA binding domains [RRMs]) with the carboxyl terminus of the HOXA9 protein (containing the homeobox domain) (*Figure 1A*). We infected murine hematopoietic stem/progenitor (ckit+, Lineage-, Sca1+ [KLS]) cells with viruses encoding BCR-ABL together with either MSI2-HOXA9 or an empty vector and tested their colony-forming ability (*Figure 1B*). Compared to BCR-ABL and empty vector (hereafter referred to as just BCR-ABL), the combination of BCR-ABL- and MSI2-HOXA9-infected cells exhibited significantly increased colony formation in primary (*Figure 1C*) and secondary plating (*Figure 1D*).

To test whether the MSI2-HOXA9 fusion confers a growth advantage in vivo, we transplanted hematopoietic stem and progenitor cells infected with BCR-ABL and MSI2-HOXA9 or an empty vector into lethally irradiated recipient mice and monitored disease development over time (*Figure 1B*). Animals transplanted with cells harboring both BCR-ABL and MSI2-HOXA9 exhibited far more aggressive disease with an ~3-fold increase in chimerism relative to the cohort receiving cells carrying BCR-ABL alone (*Figure 1E*). To determine whether the disease observed with BCR-ABL and MSI2-HOXA9 was more aggressive because it was more undifferentiated than the disease driven by BCR-ABL alone, we compared the BCR-ABL/MSI2-HOXA9 leukemic cells to the previously established model of bcCML driven by BCR-ABL/NUP98-HOXA9 (*Dash et al., 2002*). To this end, we transplanted mice with KLS cells infected with either BCR-ABL, BCR-ABL/NUP98-HOXA9, or BCR-ABL/MSI2-HOXA9 and sorted Lin- leukemia cells from these mice. Even though Lin- cells were compared, which are generally more undifferentiated, based on Giemsa staining, BCR-ABL/MSI2-HOXA9 cells displayed an increase in less differentiated blasts (82% vs. 61%) and a concomitant decrease in more differentiated cells compared to BCR-ABL-driven disease and, in this regard, was most consistent with the previously described bcCML model (*Figure 1F–I*, *Figure 1—figure supplement 1A–C*). Additionally, by FACS, total Lin- content was also increased in BCR-ABL/MSI2-HOXA9 relative to BCR-ABL alone (57% vs. 16%, *Figure 1—figure supplement 1D*). The change in differentiation state driven by MSI2-HOXA9 cells suggested that when coupled with BCR-ABL, MSI2-HOXA9 can drive chronic phase CML to blast crisis, and thus serve as an oncogene.

Reflecting the development of this more aggressive disease, mice transplanted with BCR-ABL/MSI2-HOXA9-infected cells displayed significantly reduced survival. Thus, while the median survival of mice carrying BCR-ABL alone was 37 days, that of the BCR-ABL/MSI2-HOXA9 group was just 18 days, an 8-fold increase in the risk of death (HR = 0.11) (*Figure 1J*). Collectively, these findings indicate that the MSI2/HOXA9 fusion protein has oncogenic activity and can cooperate with BCR-ABL to drive cancer progression.

## MSI2 RNA binding domains play a key role in BCR-ABL/MSI2-HOXA9-driven oncogenesis

Given that MSI2 and HOXA9 both play key roles in regulation of gene expression – MSI2 as an RNA binding protein and HOXA9 as a homeodomain transcription factor – we tested whether the oncogenic ability of MSI2-HOXA9 was dependent on the RNA binding domains of MSI2 (RRM1 and RRM2) or on the DNA binding domain (DBD) of HOXA9. We generated MSI2-HOXA9 domain mutants lacking either RRM1 or RRM2, as well as a truncated mutant lacking the HOXA9 DBD (*Figure 2A*). To test whether the domain mutants would blunt the growth advantage provided by MSI2-HOXA9 in the presence of BCR-ABL, we infected KLS cells with viruses encoding BCR-ABL and either full-length MSI2-HOXA9 or the domain mutants and performed colony assays. The loss of either RRM1 or RRM2 led to a significant loss in colony-forming ability compared to cells infected with BCR-ABL/MSI2-HOXA9 (>2-fold for both ΔRRM1 and ΔRRM2), indicating that each RNA binding domain was required for MSI2-HOXA9 activity (*Figure 2B*). While loss of the HOXA9 domain did not significantly impact primary colony formation, some impact (1.7-fold reduction) was seen at secondary plating (*Figure 1—figure supplement 1E*). These data suggest that the domains could be differentially important, with the RRMs having a dominant impact on bcCML growth. Consistent with the impact on growth in vitro, transplantation of BCR-ABL/MSI2-HOXA9ΔRRM1 (also listed as BCR-ABL/ΔRRM1) blunted leukemic growth in vivo, while loss of the HOXA9 portion of the MSI2-HOXA9 fusion did not impact chimerism compared to BCR-ABL/MSI2-HOXA9 (~30% for MSI2-HOXA9, ~25% for ΔHOXA9, ~7% for ΔRRM1; *Figure 2C*; compare with *Figure 1E*). Although each RRM was necessary for bcCML growth, expression of full-length wild-type MSI2 did not impart any growth advantage when combined with BCR-ABL in either primary (*Figure 2D*) or secondary colony assays (*Figure 2E*) relative to BCR-ABL alone. The fact that the 5′ portion of MSI2 but not the full-length protein is oncogenic suggests that the 3′ end may regulate or mask MSI2 oncogenic activity, consistent with the highly regulated activity of wild-type MSI2 proteins during development and in stem cell maintenance. Whether this regulatory effect is due to altered binding activity resulting from a change in MSI2 protein conformation, a shift in cellular localization relative to wild-type MSI2, or through some other mechanism is not clear and represents an important question for future study.

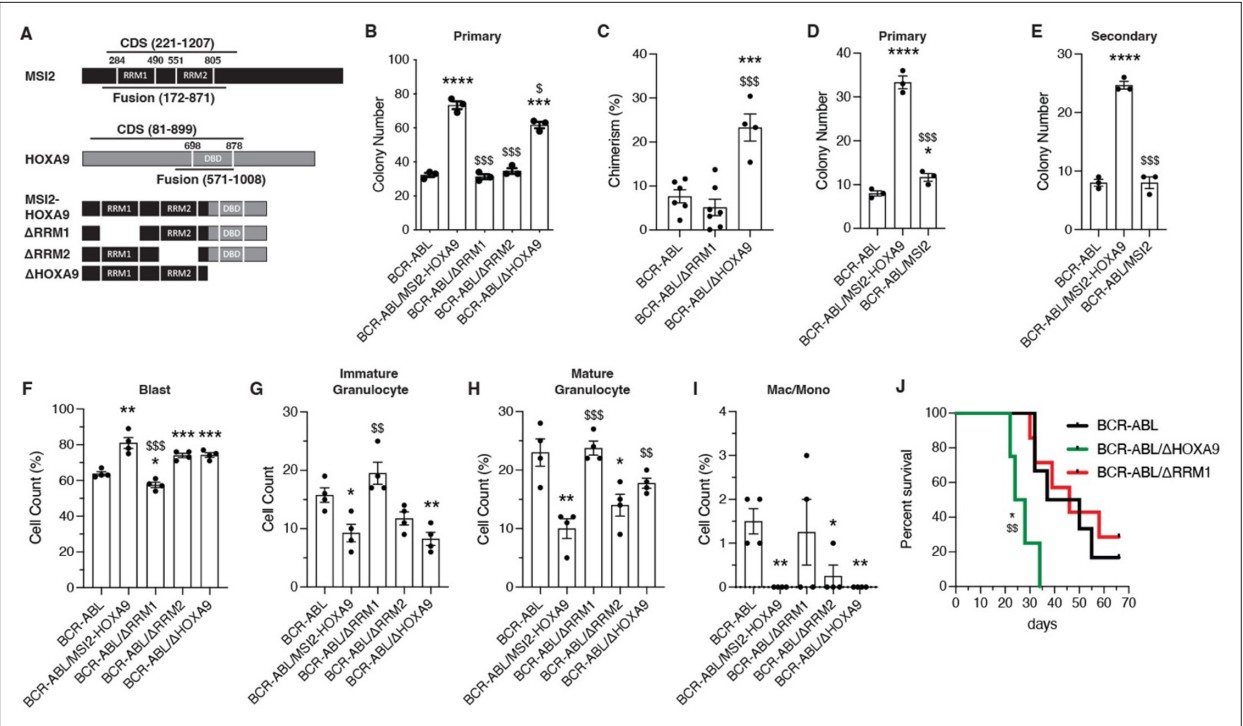

**Figure 2.** BCR-ABL/MSI2-HOXA9 cell growth and differentiation is dependent on MSI2 RRM1. (**A**) Schematic of MSI2-HOXA9 fusion protein and mutant versions generated to test domain dependencies. ΔRRM1 and ΔRRM2 are deletions of the MSI2 RNA binding domains 1 and 2, respectively, while ΔHOXA9 introduces a stop codon at the break point to eliminate the HOXA9 DNA binding domain. (**B**) Primary colony formation of KLS cells expressing BCR-ABL/Control or BCR-ABL+ variations of the MSI2-HOXA9 fusion protein. *=significance from BCR-ABL/Control, ***p=0.0002, ****p<0.0001; $=significance from BCR-ABL/MSI2-HOXA9, $p=0.01, $$$p=0.0001 for ΔRRM1, $$$p=0.0002 for ΔRRM2 (n=3 technical replicates). (**C**) Chimerism of BCR-ABL/Control, BCR-ABL/MSI2-HOXA9ΔRRM1 (also denoted as BCR-ABL/ΔRRM1), or BCR-ABL/MSI2-HOXA9ΔHOXA9 (also denoted as BCR-ABL/ΔHoxA9) cells 14 days posttransplantation. ***p=0.0009 significance from BCR-ABL/Control, $$$p=0.0004 significance from BCR-ABL/ΔRRM1 (n=6 mice for BCR-ABL/Control, n=7 for BCR-ABL/ΔRRM1, n=4 for BCR-ABL/ΔHOXA9). (**D**) Primary colony formation of KLS cells expressing BCR-ABL/Control or BCR-ABL+ MSI2-HOXA9 fusion protein or full-length wild-type MSI2. *=significance from BCR-ABL/Control, *p=0.02, ****p<0.0001; $=significance from BCR-ABL/MSI2-HOXA9, $$$p=0.0002 (n=3 technical replicates). (**E**) Secondary colony formation of KLS cells expressing BCR-ABL/Control or BCR-ABL+ MSI2-HOXA9 fusion protein or full-length wild-type MSI2. *=significance from BCR-ABL/Control, ****p<0.0001; $=significance from BCR-ABL/MSI2-HOXA9, $$$p=0.0002 (n=3 technical replicates). (**F–I**) Lin- BCR-ABL/Control or BCR-ABL+ variations of the MSI2-HOXA9 fusion protein were sorted from primary transplants and used to generate cytospins that were then stained with Giemsa and May-Grunwald stains. *=significance from BCR-ABL/Control, $=significance from BCR-ABL/MSI2-HOXA9 (n=4 mice per group). (**F**) Quantification of blast cells. *p=0.01, **p=0.001, ***p=0.0008, $$$p=0.0004. (**G**) Quantification of immature granulocytes. *p=0.01, **p=0.004, $$p=0.005. (**H**) Quantification of mature granulocytes. *p=0.02, **p=0.004, $$p=0.006, $$$p=0.0005. (**I**) Quantification of differentiated macrophages and monocytes. *p=0.01, **p=0.002. (**J**) Survival of mice transplanted with BCR-ABL/Control (n=5 mice), BCR-ABL/ΔRRM1 (n=6 mice), or BCR-ABL/ΔHOXA9 (n= 4 mice) expressing KLS cells. *p=0.01 significance from BCR-ABL/Control, $$p=0.007 significance from BCR-ABL/ΔRRM1. Two-tailed unpaired Student's t-tests were used to determine statistical significance.

We also tested which domains may be important for driving the more undifferentiated state created by MSI2-HOXA9. Analysis of established primary cancer cells stained with Giemsa and May-Grunwald stains indicated that while both RRM1 and RRM2 similarly impacted growth in colony assays (78 colonies for MSI2-HOXA9, ~35 colonies for both ΔRRM1 and ΔRRM2; *Figure 2B*, *Figure 1—figure supplement 1E*), only deletion of RRM1 impacted differentiation (*Figure 2F–I*, *Figure 1—figure supplement 1F–H*), leading to a reduction in blast cell counts (~80% for MSI2-HOXA9, 60% for BCR-ABL only, 58% for ΔRRM1; *Figure 2F*) and an increase in differentiated myeloid cells (*Figure 2G–I*), and resulting in a more chronic phase disease similar in composition to that driven by BCR-ABL alone.

In comparing the two MSI2 RRMs, the data above suggested that RRM1 is more critical than RRM2. This is consistent with RRM1 having been shown to provide most of the specificity and higher binding energy relative to RRM2 (*Ohyama et al., 2012*), and thus we focused on RRM1 deletion for subsequent in vivo investigation. Further, supporting the observation that the MSI2 RRM1 is critical for growth and differentiation, the deletion of RRM1 from Msi2-HoxA9 led to significantly longer survival that was

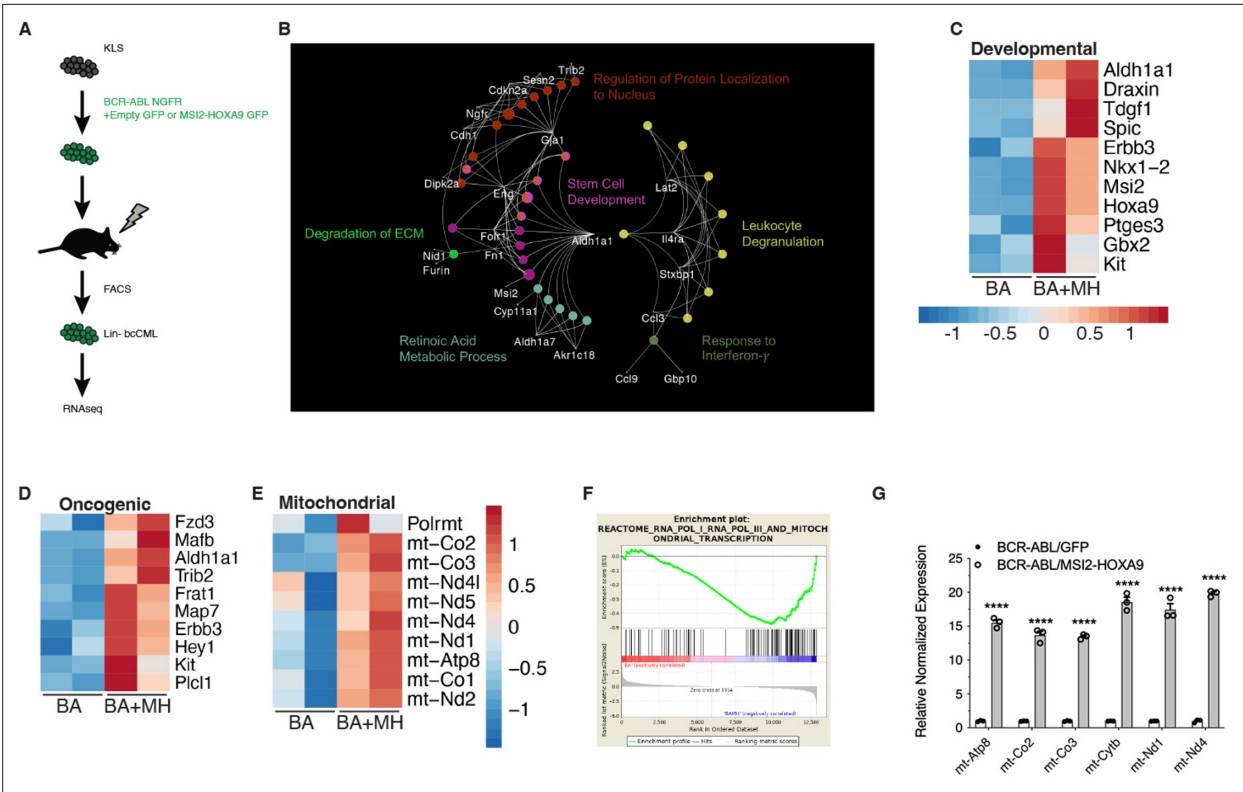

**Figure 3.** MSI2-HOXA9 regulates gene expression, including mitochondrial, developmental, and oncogenic genes. (**A**) Schematic of RNA-seq workflow. (**B**) Network map of differential genes from RNA-seq of Lin- BCR-ABL/Control vs. Lin- BCR-ABL/MSI2-HOXA9. (**C**) Heatmap of developmental genes from RNA-seq of Lin- BCR-ABL/Control vs. Lin- BCR-ABL/MSI2-HOXA9. (**D**) Heatmap of oncogenic genes from RNA-seq of Lin- BCR-ABL/Control vs. Lin- BCR-ABL/MSI2-HOXA9. (**E**) Heatmap of mitochondrial genes from RNA-seq of Lin- BCR-ABL/Control vs. Lin- BCR-ABL/MSI2-HOXA9. (**F**) Gene set enrichment analysis plot of mitochondrial transcription program generated from RNA-seq of Lin- BCR-ABL/Control vs. Lin- BCR-ABL/MSI2-HOXA9. (**G**) Validation of selected mitochondrial genes upregulated in Lin- BCR-ABL/MSI2-HOXA9 relative to Lin- BCR-ABL/Control. ****p<0.0001 (n=3 technical replicates). Two-tailed unpaired Student's t-tests were used to determine statistical significance.

The online version of this article includes the following figure supplement(s) for figure 3:

**Figure supplement 1.** Principal component analysis and mitochondrial parameters associated with BCR-ABL/MSI2-HOXA9 leukemia.

no different from the survival of mice transplanted with BCR-ABL alone (p=0.97) (*Figure 2J*). Thus, while the median survival of mice with leukemia driven by BCR-ABL/MSI2-HOXA9ΔHOXA9 (BCR-ABL/ΔHOXA9) was 26 days, the loss of RRM1 led to a median survival of 46 days, similar to BCR-ABL alone (43.5 days). This resulted in a 12-fold (HR = 0.08) increase in risk of death for BCR-ABL/ΔHOXA9 compared to BCR-ABL only and a 15-fold (HE = 0.06) increase in risk of death for BCR-ABL/ΔHOXA9 compared to BCR-ABL/ΔRRM1. Together, these results demonstrate that the MSI2 RRM1 domain is preferentially critical for the lethality seen in BCR-ABL/MSI2-HOXA9 bcCML .

## MSI2-HOXA9- triggers global changes in transcriptional programs

To understand how the *MSI2-HOXA9* translocation promotes bcCML, we compared gene expression patterns in BCR-ABL and BCR-ABL/MSI2-HOXA9-driven disease. RNA-seq analysis was carried out on lineage-negative cells from leukemia established with BCR-ABL or BCR-ABL/MSI2-HOXA9 (*Figure 3A*, *Figure 3—figure supplement 1A*). Network mapping of all differentially expressed genes (q-value<0.05) using nonredundant functional grouping revealed an enrichment of metabolic processes with oncogenic pathways and developmental programs (*Figure 3B*). Programs that were dominantly upregulated by MSI2/HOXA9 were those involved in development, including *Aldh1a1*, *Erbb3*, and *Kit*, consistent with BCR-ABL/MSI2-HOXA9 driving a more undifferentiated disease (*Figure 3C*), known oncogenes, including *Frat1*, *Map7*, and *Fzd3* (*Figure 3D*), and components of the mitochondria, including *mt-Co2*, *mt-Atp8*, and *mt-Nd5* (*Figure 3E*). Gene set enrichment analysis (GSEA) suggested an enrichment in genes associated with the mitochondrial transcription pathway

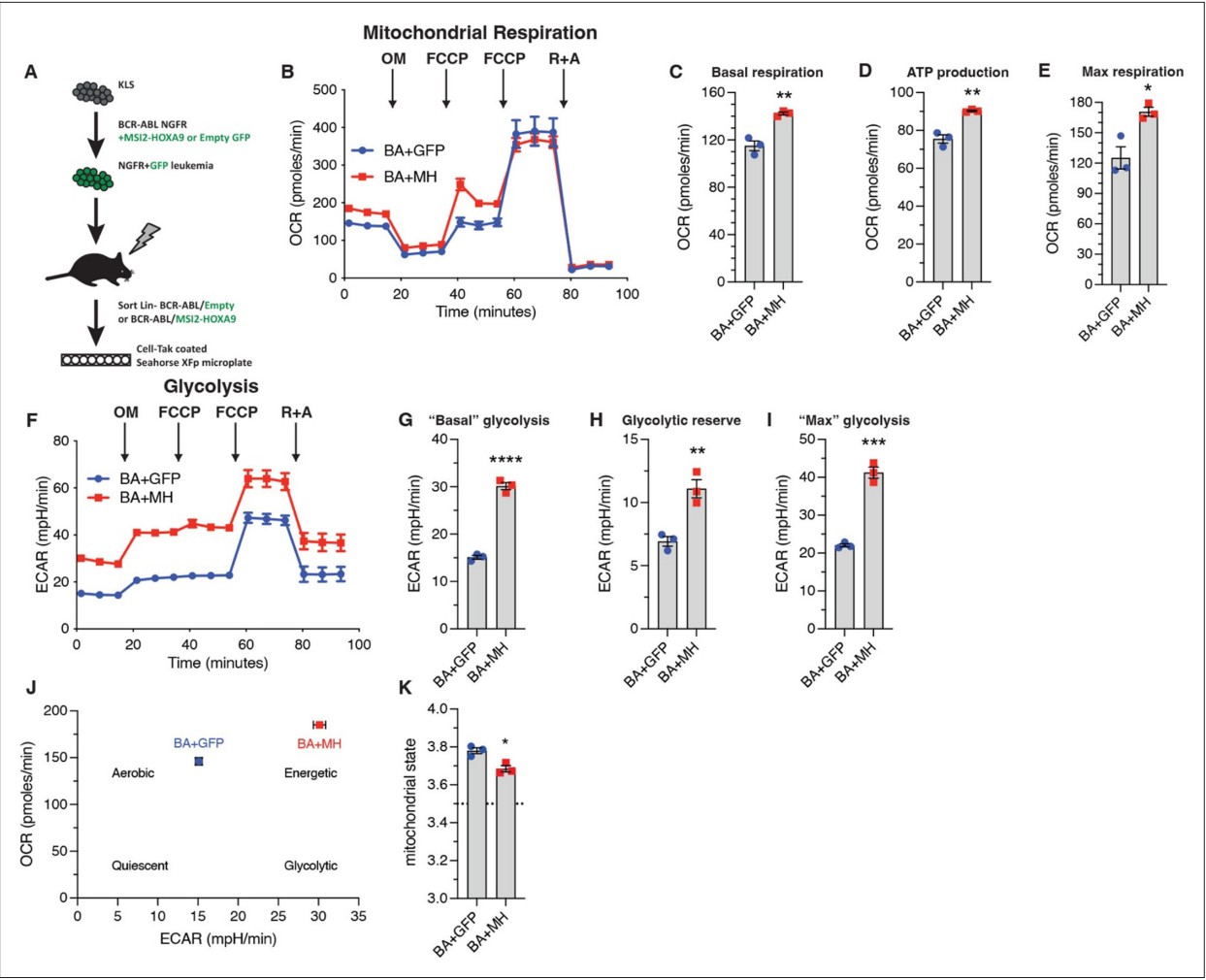

**Figure 4.** Elevated mitochondrial gene expression leads to increased cellular respiration and energetics. (**A**) Schematic of experimental workflow to assess Lin- BCR-ABL/Control vs. Lin- BCR-ABL/MSI2-HOXA9 cancer cell energetics using the Seahorse XFp Cell Mito Stress Test Kit. (**B**) Oxygen consumption rate (OCR) of Lin- BCR-ABL/Control and Lin- BCR-ABL/MSI2-HOXA9 to measure mitochondrial respiration (n=3 technical replicates for each cell type). (**C**) Basal respiration rate of Lin- BCR-ABL/Control and Lin- BCR-ABL/MSI2-HOXA9 determined from the OCR trace. **p=0.003 (n=3 technical replicates for each cell type). (**D**) ATP production rate of Lin- BCR-ABL/Control and Lin- BCR-ABL/MSI2-HOXA9 determined from the OCR trace. **p=0.002 (n=3 technical replicates for each cell type). (**E**) Maximum respiration rate of Lin- BCR-ABL/Control and Lin- BCR-ABL/MSI2-HOXA9 determined from the OCR trace after the first FCCP injection. *p=0.01 (n=3 technical replicates for each cell type). (**F**) Extracellular acidification rate (ECAR) of Lin- BCR-ABL/Control and Lin- BCR-ABL/MSI2-HOXA9 to measure glycolysis (n=3 technical replicates for each cell type). (**G**) 'Basal' glycolysis of Lin- BCR-ABL/Control and Lin- BCR-ABL/MSI2-HOXA9 determined from the ECAR trace, ****p<0.0001 (n=3 technical replicates for each cell type). (**H**) Glycolytic reserve of Lin- BCR-ABL/Control and Lin- BCR-ABL/MSI2-HOXA9 determined from the ECAR trace, **p=0.006 (n=3 technical replicates for each cell type). (**I**) 'Maximum' glycolysis of Lin- BCR-ABL/Control and Lin- BCR-ABL/MSI2-HOXA9 determined from the ECAR trace, ***p=0.0002 (n=3 technical replicates for each cell type). (**J**) Energetic landscape of Lin- BCR-ABL/Control and Lin- BCR-ABL/MSI2-HOXA9 generated by plotting the basal OCR vs. basal ECAR for each cell type. (**K**) Mitochondrial state determined from OCR data. *p=0.01 (n=3 technical replicates for each cell type). Two-tailed unpaired Student's t-tests were used to determine statistical significance.

(p-value<0.05) (*Figure 3F*); this finding was subsequently confirmed by qRT-PCR (*Figure 3G*). The impact on mitochondrial gene expression was unexpected, and we thus pursued this to gain insight into whether BCR-ABL/MSI2-HOXA9-driven mitochondrial changes may be pivotal in driving leukemia growth.

Among the genes most enriched in BCR-ABL/MSI2-HOXA9-driven bcCML were those encoding key components of the mitochondrial respiratory chain, including Complex I, Complex III, Complex IV, and the F0 subunit of ATP synthase (*mt-Atp6* and *mt-Atp8*). Analysis of metabolic capacity and mitochondrial function on Lin- bcCML using the Seahorse Cell Mito Stress Test (*Figure 4A*) revealed that MSI2-HOXA9 significantly increased basal respiration (*Figure 4B and C*), ATP production

(*Figure 4D*), and maximal respiration (*Figure 4E*). These three findings suggest that the introduction of MSI2-HOXA9 results in a cell operating at a higher energetic level. In contrast, most of the other mitochondrial parameters we evaluated were not significantly impacted (*Figure 3—figure supplement 1B–E*). Further, BCR-ABL/MSI-HOXA9 increased basal glycolysis (*Figure 4F and G*), glycolytic reserves (*Figure 4H*), and maximum glycolysis (*Figure 4I*). Again, these data suggest that in addition to requiring a higher basal energy level, they also exhibit greater maximal energy capacity. We also observed an increase in the lactate output to glucose uptake ratio in BCR-ABL/MSI2-HOXA9 cells compared to BCR-ABL cells in vitro (*Figure 3—figure supplement 1F–H*), suggesting greater energy demands in these cells. A plot of the basal oxygen consumption rate (OCR) against extracellular acidification rate revealed a cellular energetics landscape wherein BCR-ABL/MSI2-HOXA9 cells are characteristically more energetic than BCR-ABL cells (*Figure 4J*). Finally, OCR data showed that BCR-ABL/MSI2-HOXA9 cells operate in a more energetically optimal way; their mitochondria were in a state close to 3.5 (*Figure 4K*), with state 3 being the theoretical maximal respiration state (*Chance and Williams, 1955*). Collectively, these findings suggest that MSI2-HOXA9 exerts its effects at least in part by augmenting mitochondrial function and basal respiration in leukemia cells.

## MSI2-HOXA9 impacts mitochondrial gene expression through control of the mitochondrial polymerase, POLRMT

To understand the basis of the alteration in mitochondrial gene expression in BCR-ABL/MSI2-HOXA9 leukemia and the concomitant increase in energy production, we tested whether MSI2-HOXA9 impacted mitochondrial gene expression directly in the mitochondria or indirectly through control of the polymerase and/or transcription factors. We thus assessed expression of the mitochondrial polymerase and selected mitochondrial transcription factors and found a 5-fold increase in expression

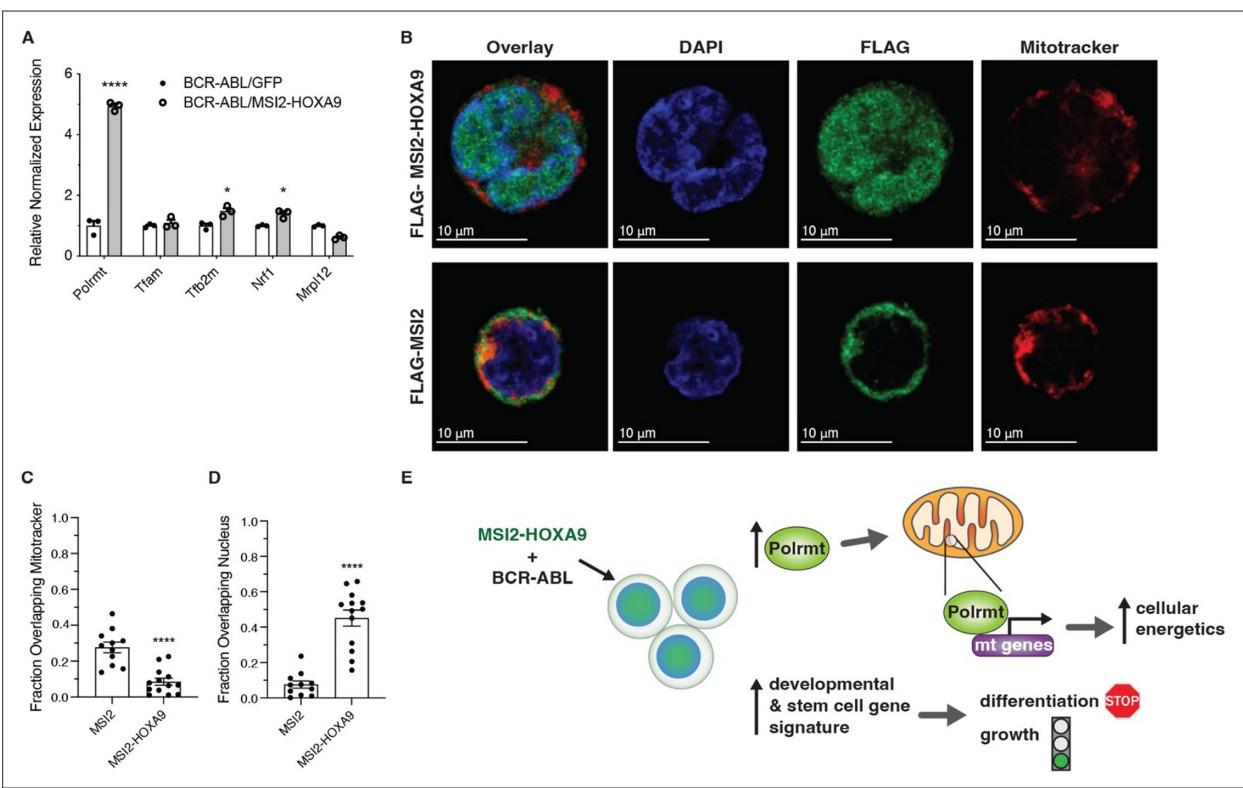

**Figure 5.** MSI2-HOXA9 localizes to the nucleus and increases Polrmt expression. (**A**) qRT-PCR of genes directly involved in regulation of mitochondrial gene expression. *p=0.02 for *Tfb2m*, *p=0.01 for *Nrf1*, ****p<0.0001 for *Polrmt* (n=3 technical replicates). (**B**) Representative immunofluorescent images of K562 cells expressing Flag-tagged MSI2 or Flag-tagged MSI2-HOXA9 (scale bar = 10 μm). (DAPI = blue, FLAG-MSI2 or FLAG-MSI2-HOXA9=green, mitotracker = red). (**C**) Overlap of Flag-tagged MSI2 or MSI2-HOXA9 with mitochondria. ****p<0.0001 (n=11 for MSI2, n=13 for MSI2-HOXA9). (**D**) Overlap of Flag-tagged MSI2 or MSI2-HOXA9 with the nucleus. ****p<0.0001 (n=11 for MSI2, n=13 for MSI2-HOXA9). (**E**) Proposed mechanism by which the MSI2-HOXA9 fusion protein promotes an undifferentiated aggressive cancer cell. Two-tailed unpaired Student's t-tests were used to determine statistical significance.

of the lone mitochondrial polymerase (*Polrmt*) in BCR-ABL/MSI2-HOXA9 cells, as well as significant increases in the transcription initiation factor *Tfb2m* and the mitochondrial transcription factor regulator *Nrf1*, 1.5-fold and 1.4-fold, respectively (*Figure 5A*). While *Tfb2m* and *Nrf* showed a significant increase in expression, we chose to focus on the much greater impact seen on *Polrmt* expression. Since these components are transcribed in the nucleus, the data suggest that MSI2-HOXA9 controls mitochondrial gene expression by promoting expression of POLRMT and key transcription factors. Consistent with this, MSI2-HOXA9 was preferentially found in the nucleus relative to MSI2 (*Figure 5B–D*). Further, we identified 10 occurrences of the minimal MSI2 RRM1 consensus binding sequence UAGU within the *Polrmt* transcript. These data suggest a model in which MSI2-HOXA9 localizes to the nucleus and elevates *Polrmt* expression. POLRMT and the mitochondrial transcription factors, in turn, control expression of mitochondrial genes that encode the components of the respiratory chain, leading to increased mitochondrial function and cellular energetics that drive the aggressiveness of bcCML (*Figure 5E*).

## Discussion

The stem cell gene *MSI2* is often hijacked and aberrantly upregulated in cancer, where it can drive sustained self-renewal of cancer stem cells, leading to tumor growth, therapy resistance, and disease progression. Although MSI2 was shown to be a dependency of aggressive myeloid leukemias, as well as solid cancers such as pancreatic cancer (*Fox et al., 2016*; *Ito et al., 2010*; *Kwon et al., 2015*; *Kharas et al., 2010*), and its transgenic overexpression can lead to increased tumor burden (*Kharas et al., 2010*), whether *MSI2* translocations or mutations harbored in patients can serve as oncogenes and driver mutations remains unknown. Because a translocation between *MSI2* and *HOXA9* had been reported to occur in bcCML patients, we focused on testing whether this translocation could be an oncogene in myeloid leukemia. Our data presented here show that overexpression of the *MSI2-HOXA9* allele together with *BCR-ABL* can trigger the development of bcCML. These findings are important in that they represent the first demonstration that an aberrant *MSI2* genetic lesion found in patients can be oncogenic.

The new bcCML mouse model we developed is a valuable resource to the field, particularly because there is only one other model for bcCML (*Dash et al., 2002*) developed over 20 years ago. bcCML is an aggressive, highly lethal malignancy, and new models are critical to better understand the biology of the disease, identify molecular dependencies, and test new approaches to therapy. The BCR-ABL/MSI2-HOXA9 mouse model is characterized by a shift in cellular composition during disease progression, with an increase in blast cells and a decrease in more differentiated blood cells. This is consistent with the human disease, as well as with the existing NUP98-HOXA9/BCR-ABL mouse model. Importantly, comparison of MSI2-HOXA9 and NUP98-HOXA9-driven leukemias may reveal activation of shared, as well as unique programs. Analysis in each case is valuable: identification of common pathways activated in the two models may reveal underlying programs that are fundamental to bcCML growth, regardless of driver gene, while identification of unique pathways may provide key insight into mechanisms of initiation and identify oncogene-specific dependencies. Overall, the MSI2-HOXA9 leukemia model we developed will contribute to a deeper understanding of how a benign, chronic disease transitions to an aggressive, highly lethal disease and may reveal new vulnerabilities that can be targeted to stop this deadly progression.

Domain mutants of the MSI2-HOXA9 protein allowed us to better delineate the contributions of the two RRM domains and the DBD domain of HOXA9. Deletion of the MSI2 RRM1 reduced the oncogenicity of the MSI2-HOXA9 fusion in vitro and in vivo, resulting in a disease similar to BCR-ABL alone as chimerism and survival were similar to the CML model. RRM1 deletion prevented the increase in blast cells that the full fusion produced and resulted in a composition of differentiated cells and blasts similar to BCR-ABL. While the loss of the HOXA9 portion of the MSI2-HOXA9 protein had some impact in secondary plating in vitro, loss of the HOXA9 portion did not impact survival and progression of bcCML in vivo. These data thus identify the RRM1 domain of the new allele as the predominant driver of MSI-HOXA9's oncogenic capacity.

To understand the downstream mechanism by which MSI2-HOXA9 cooperated with BCR-ABL to drive CML into blast crisis, we analyzed the transcriptomic changes triggered by MSI2-HOXA9 in the context of BCR-ABL expression. Among the global changes in gene expression patterns observed, the most highly enriched factors were developmental and stem cell signals such as *Aldh1a1*, *Erbb3*,

and *Kit*, consistent with the undifferentiated nature of bcCML that was driven by MSI2-HOXA9. In addition, there was a striking increase in the expression of mitochondrial genes, with many critical components of the respiratory chain significantly elevated. This translated to increased mitochondrial activity, as well as an increase in basal respiration and cellular energetics. Among the genes regulated through MSI2-HOXA9 activity was *Polrmt*, the sole RNA polymerase that controls mitochondrial gene expression. Our data suggest that dysregulated POLRMT activity upregulates the expression of key respiratory chain proteins, increasing mitochondrial function and helping to fuel uncontrolled growth of the leukemia. While the increased mitochondrial energetics may be related to the increased proliferation that is also a hallmark of bcCML, several studies have suggested a link between mitochondrial activity and leukemia cell differentiation as well. For example, *Yehudai et al., 2019* showed that blocking mitochondrial DNA replication using either genetic knockdown or chemical inhibitors promotes the differentiation of human AML cells. Further, in a study examining the relationship between microvesicle-driven release of mitochondria, energy metabolism, and leukemia cell differentiation, *Zhao et al., 2020* showed that disrupting the release of mitochondria or blocking mitochondrial function significantly impacts the differentiation of human leukemia cell lines.

The possibility that a fundamental relationship exists between mitochondrial pathways and the differentiation state of aggressive leukemias is exciting and may reveal new strategies for therapy. The contribution of mitochondrial activity to proliferation versus differentiation in blast crisis CML driven by BCR-ABL /MSI2-HOXA9 will be an important question to address in future work.

While the main focus of the work reported here is the role of *MSI2* translocation in hematologic malignancies, this discovery may lay a foundation for defining the role *MSI2* genetic lesions may play as oncogenes in other disease settings. Not only have other *MSI2* fusions been detected in patient leukemia samples (i.e. *PAX5-MSI2* [*Wang et al., 2017*], *EVI1-MSI2* [*De Weer et al., 2008*], *TTC40-MSI2* [*Saleki et al., 2015*]), but we have found that the *MSI2* gene is also a frequent partner (in both the 3 prime and 5 prime locations) for an array of translocations in solid cancers, including breast, lung, and colon cancer (*Jang et al., 2020*; *Supplementary file 1*). This suggests that *MSI2* could be functioning as an oncogene in human disease in multiple other contexts, and with multiple other partners. Thus, our work demonstrating that *MSI2* translocations can be oncogenic in the context of bcCML may open new avenues and provide a paradigm for understanding how dysregulation of this pathway may be oncogenic in a wider array of cancers.

## Materials and methods

### Mice
All animal experiments were performed according to protocols approved by the University of California San Diego Institutional Animal Care and Use Committee (#S10274). Mice were bred and maintained in the animal care facilities at the University of California San Diego. The following mice were used: B6-CD45.2 and B6-CD45.1 (Strain: B6.SJL-Ptprc$^a$Pepc$^b$/BoyJ); NSG mice (Strain: NOD.Cg-Prkdc$^{scid}$ Il2rg$^{tm1Wjl}$/SzJ). All mice were 8–16 weeks of age.

### Generation and analysis of leukemic mice
For BCR-ABL1/NUP98-HOXA9-driven bcCML, BCR-ABL-driven CML, or BCR-ABL/MSI2-HOXA9, bone marrow-derived KLS cells were isolated and sorted from CD45.2 B6 mice. All sorted cells were cultured overnight in X-Vivo15 media (Lonza) supplemented with 50 µM 2-mercaptoethanol, 10% (vol/vol) fetal bovine serum (FBS), 1% penicillin-streptomycin, SCF (100 ng/ml, R&D Systems) and TPO (20 ng/ml, R&D Systems). Cells were retrovirally infected with MSCV-BCR-ABL-IRES-NGFR and MSCV-NUP98-HOXA9-IRES-huCD2 to generate bcCML; MSCV-BCR-ABL-IRES-NGFR and MSCV-EMPTY-IRES-GFP to generate CML; MSCV-BCR-ABL-IRES-NGFR and MSCV-MSI2-HOXA9-IRES-GFP to generate BCR-ABL/MSI2-HOXA9 leukemia. Subsequently, cells were harvested 48 hr after infection. For primary transplants, infected and sorted cells were transplanted retro-orbitally into cohorts of sublethally (6 Gy) irradiated CD45.1 mice. Disease mice were analyzed as previously described (*Zimdahl et al., 2014*).

## In vitro methylcellulose colony formation assay

For primary plating, KLS cells isolated from B6-CD45.2 mice were infected as described above, incubated in the presence of virus for 48 hr, and then sorted for oncogene tags (NGFR and GFP). 250 cells were plated in triplicate in cytokine-free methylcellulose media (Methocult GM M3234, StemCell Technologies) supplemented with 20 ng/ml IL-3 and 20 ng/ml SCF (R&D Systems), 2% FBS, and 1% penicillin-streptomycin. For secondary plating, primary plates were dissociated, replicates were combined, and cells were collected and washed in PBS. 2500 cells were re-plated in fresh methylcellulose media and cytokines.

## Cell isolation and FACS analysis

Cells were suspended in Hanks' balanced salt solution (Gibco, Life Technologies) containing 5% (vol/vol) FBS and 2 mM EDTA and prepared for FACS analysis and sorting as previously described (*Domen et al., 2000*). Red blood cells were lysed using RBC Lysis Buffer (eBioscience) before antibody incubation. For KLS isolation, cells were incubated with CD117 (ckit) magnetic beads (Miltenyi Biotec) followed by positive selection using an AutomacsPro (Miltenyi Biotec). Following ckit enrichment, cells were incubated with 2B8 (ckit), D7 (Sca1) and the following antibodies to define lineage-positive cells: 145–2C11 (CD3ε), GK1.5 (CD4), 53–6.7 (CD8), RB6-8C5 (Ly-6G/Gr1), M1/70 (CD11b/Mac-1), TER119 (Ly-76/TER119), 6B2 (CD45R/B220), and MB19-1 (CD19). The same general procedure of RBC lysis and antibody incubation was used for all other flow cytometry experiments. All antibodies were purchased from BD Pharmingen, eBioscience, or BioLegend. All cell sorting and flow cytometry analysis was carried out on a FACSAria III machine (Becton Dickinson), and data were analyzed with FlowJo software (version 10.7.1).

## Immunofluorescence staining for mitochondria

The day prior to plating cells, four-well removable chamber slides (Lab Tek II) were coated with Cell-Tak and kept at 4°C overnight. The next day, FLAG-tag MSCV-MSI2-IRES-GFP or FLAG-tag MSCV-MSI2HOXA9-IRES-GFP-infected K562 cells were labeled with 100 nM MitoTracker Deep Red FM (Thermo Fisher) at 37°C. After washing twice with PBS, cells were fixed with 4% PFA at 37°C and washed with warm PBS. Cells were spun onto the Cell-Tak-coated chamber slides with the brake off and then allowed to settle for an additional 30 min. Cells were then permeabilized with 0.1% Triton X-100 for 3 min and then blocked in PBS with 10% normal goat serum, 5% bovine serum albumin, 0.3 M glycine, and 0.05% Tween-20 for 1 hr. After blocking, cells were incubated with primary antibody in 1:10 diluted blocking buffer overnight at 4°C. The following primary antibodies were used: chicken anti-GFP 1.5:1000 (Abcam) and rabbit anti-FLAG 1.5:1000 (Cell Signaling Technology). Alexa Fluor-conjugated secondary antibody (1:500) incubation was performed for 1 hr at room temperature. DAPI (4',6-diamidino-2-phenylindole; Molecular Probes) was used to detect DNA. Images were obtained with a Zeiss LSM-700 confocal microscope.

## Giemsa and May-Grunwald staining

Lin- cancer was sorted from transplanted mice and spun onto cytospin slides. Slides were allowed to air dry overnight before staining with May-Grunwald (Sigma) and Giemsa (diluted 1:20 with DI H$_2$O, Sigma) stains and again allowed to air dry overnight. Imaging was done using the Keyence BZX-700 fluorescent microscope.

## Retroviral constructs and production

MIG-BCR-ABL was provided by Warren Pear and Ann Marie Pendergast and was cloned into the MSCV-IRES-NGFR retroviral vector. MSCV-NUP98-HOXA9-IRES-YFP was provided by Gary Gilliland and was subcloned into the MSCV-IRES- huCD2 vector (or MSCV-IRES-GFP) retroviral vector. Virus was produced in 293T cells transfected using X-tremeGENE HP (Roche) with viral constructs along with VSV-G and gag-pol. Viral supernatants were collected for 3 days followed by ultracentrifugal concentration at 20,000 × *g* for 2 hr.

## qRT-PCR analysis

RNA was isolated using RNeasy Micro Kit (QIAGEN), and RNA was converted to cDNA using Superscript III reverse transcriptase (Life Technologies). Quantitative real-time PCRs were performed

using an iCycler (Bio-Rad) by mixing cDNAs, iQ SYBRGreen Supermix (Bio-Rad), and gene-specific primers. Gene expression was normalized to the levels of Beta-2 microglobulin (*B2m*). All primers were designed using NCBI's Primer-BLAST (https://www.ncbi.nlm.nih.gov/tools/primer-blast/). Primer sequences are listed in *Supplementary file 2*.

## YSI bioanalyzer assay

Lin- leukemic cells were sorted from mice transplanted with BCR-ABL/Control or BCR-ABL/MSI2-HOXA9-transduced KLS cells. Sorted cells were plated in round-bottom plates in X-Vivo15 media (Lonza) supplemented with 50 µM 2-mercaptoethanol, 10% (vol/vol) FBS, 1% penicillin-streptomycin, SCF (100 ng/ml, R&D Systems) and TPO (20 ng/ml, R&D Systems). The media supernatant was collected on days 1, 2, and 3, and cell counts were obtained by Trypan Blue counting on days 1 and 3. Glucose, glutamine, glutamate, and lactate concentration in the collected supernatant were measured using a YSI 2950 metabolite analyzer.

## Seahorse assay

The day prior to running the assay, the wells of a XFp plate were coated with Cell-Tak and the plate was kept at 4°C overnight. Lin- leukemic cells were sorted from mice transplanted with BCR-ABL/Control or BCR-ABL/MSI2-HOXA9-transduced KLS cells. Sorted cells were washed twice with complete assay media: XF-DMEM pH 7.4, 1 mM pyruvate, 2 mM L-glutamine, and 10 mM D-glucose. The cells were then aliquoted, spun down, resuspended in 50 µl complete assay media, and added to the previously coated XFp plate. The XFp plate was then spun at 300×*g* for 1 min with the brake off, and cell sedimentation was observed by microscope before loading onto the Seahorse XFp analyzer.

## RNA isolation and RNA-seq analysis

Lin- leukemic cells were sorted from mice transplanted with BCR-ABL/Control or BCR-ABL/MSI2-HOXA9-transduced KLS cells. Total RNA was isolated using the RNeasy Micro Plus Kit (QIAGEN). RNA libraries were generated from 150 ng of RNA using Illumina's TruSeq Stranded mRNA Sample Prep Kit (Illumina). Libraries were pooled and single-end sequenced (1×75) on the Illumina NextSeq 500 using the High Output V2 Kit (Illumina).

BCR-ABL/Control and BCR-ABL/MSI2-HOXA9 fastq files were processed into transcript-level summaries using kallisto (*Bray et al., 2016*). Transcript-level summaries were processed into gene-level summaries, and differential gene expression was performed using sleuth with the Wald test (*Pimentel et al., 2017*). GSEA was performed as previously described (*Lytle et al., 2019*). ClueGO was used for gene enrichment analysis of all differentially expressed genes (FDR<0.05) identified between BCR-ABL/Control and BCR-ABL/MSI2-HOXA9 mouse cells. GO and Reactome gene sets were used with medium network specificity, a p-value cutoff of <0.05 and a kappa score of 0.4. All other statistical parameters remained on default settings. CluePedia was used to identify genes found within enriched gene sets. All network analyses ran and were visualized in Cytoscape 3.7 (Cytoscape, ClueGO/CluePedia).

## Cell lines

K562 and 293T cells were obtained directly from ATCC. Quality control through ATCC included testing for bacterial, fungal, and mycoplasma contamination. ATCC also performed STR profiling to authenticate the identity of the lines.

## Statistics and reproducibility

Statistical analyses were carried out using GraphPad Prism software version 9.2 (GraphPad Software Inc). All data are shown as mean ± SEM. Two-tailed unpaired Student's t-tests were used to determine statistical significance. No statistical method was used to predetermine sample size and no data were excluded from the analysis. All experiments were reproducible.

## Acknowledgements

We would like to thank Warren Pear and Ann Marie Pendergast for providing MIG-BCR-ABL and Gary Gilliland for providing MSCV-NUP98-HOXA9-IRES-YFP. We would like to thank Hannah Pettit for uploading the RNA-seq expression data to Dryad. We would also like to thank the Sanford Burnham

Prebys Cancer Metabolism Core and Dr. David Scott for their advice and contributions to the work, as well as the University of California San Diego Microscopy Core, which is supported by NIH grant NS047101. KS received support from NIH grants T32 CA009523 and T32 HL086344. MH received support from T32 HL086344. JB was supported by a postdoctoral fellowship from the National Cancer Center and an ASH Scholar Award. ED received support from T32 GM007752. This work was supported by NIH grant R35 CA197699 to TR.

## Additional information

### Funding

| Funder | Grant reference number | Author |
|---|---|---|
| National Institutes of Health | T32 CA009523 | Kyle Spinler |
| National Institutes of Health | T32 HL086344 | Kyle Spinler Michael Hamilton |
| National Cancer Center | | Jeevisha Bajaj |
| ASH Foundation | | Jeevisha Bajaj |
| National Institutes of Health | T32 GM007752 | Emily Diaz |
| National Institutes of Health | R35 CA197699 | Tannishtha Reya |

The funders had no role in study design, data collection and interpretation, or the decision to submit the work for publication.

### Author contributions

Kyle Spinler, Conceptualization, Data curation, Formal analysis, Funding acquisition, Investigation, Methodology, Writing – original draft, Writing – review and editing; Michael Hamilton, Formal analysis, Funding acquisition; Jeevisha Bajaj, Data curation, Funding acquisition, Investigation, Methodology; Yutaka Shima, Data curation, Formal analysis, Investigation; Emily Diaz, Data curation, Funding acquisition, Investigation; Marcie Kritzik, Writing – original draft, Writing – review and editing; Tannishtha Reya, Conceptualization, Resources, Formal analysis, Supervision, Funding acquisition, Writing – original draft, Writing – review and editing

### Author ORCIDs

Tannishtha Reya (iD) https://orcid.org/0000-0002-5956-8536

### Ethics
All animal experiments were performed according to protocols approved by the University of California San Diego Institutional Animal Care and Use Committee (#S10274).

Reviewer #1 (Public Review): https://doi.org/10.7554/eLife.93645.2.sa1
Reviewer #2 (Public Review): https://doi.org/10.7554/eLife.93645.2.sa2
Author response https://doi.org/10.7554/eLife.93645.2.sa3

## Additional files

### Supplementary files
MDAR checklist

Supplementary file 1. Selected MSI2 Fusions in a Broad Array of Cancers.

Supplementary file 2. List of Primer Sequences.

Supplementary file 3. List of Antibodies Used.

Supplementary file 4. Source Data for All Numerical Values Reported in Figures and Figure Supplements.

### Data availability

Primer information is provided in *Supplementary file 2*. Antibody information is provided in *Supplementary file 3*. Source data are provided with this paper as a separate Source Data file (*Supplementary file 4*). Examples of flow cytometry gating strategies are provided in *Figure 1—figure supplement 2*. Following the lab's move to Columbia University, the raw sequencing data for this manuscript could not be located; as a result, the data are not available in that form. However, complete and rigorous downstream analyses were completed using Sleuth to obtain TPM values, which have been preserved and deposited in the Dryad Digital Repository (DOI: 10.5061/dryad.sbcc2frm6). Extensive quality control and analysis originally conducted at the time of sequencing confirm the integrity and reproducibility of the provided data. Associated sequencing run metadata are also included in the Dryad submission. All other data supporting the findings of this study are available within the article and its supplementary information files.

The following dataset was generated:

| Author(s) | Year | Dataset title | Dataset URL | Database and Identifier |
|---|---|---|---|---|
| Spinler K, Hamilton M, Pettit H, Reya T | 2025 | Bulk RNA sequencing analysis of Lin- leukemia BCR-ABL and BCR-ABL/MSI2-HOXA9 cells (post-transplantation) | https://doi.org/10.5061/dryad.sbcc2frm6 | Dryad Digital Repository, 10.5061/dryad.sbcc2frm6 |

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
