## [Editor Report · eLife assessment]

The study presents **important** findings on the role of MSI2-HOXA9 translocation in chronic myeloid leukemia. The authors provide **convincing** evidence supporting the role of this translocation in leukemogenesis by using elegant mouse modeling and in vitro mechanistic studies. Consistent with the reviews, the studies can be strengthened with further murine and cell line experiments. Raw sequencing data for this manuscript are not available as the authors could not locate the files after moving institutions. Expression data for the RNA Seq data are available.

---

## [Referee Report · Reviewer #1 (Public Review)]

This is a very interesting study by Kyle Spinler et al., demonstrating the novel role of MSI2-HOXA9 translocation in the development and pathogenesis of blast crisis CML. The authors employed appropriate in vitro and in vivo assays, including a sophisticated transplantation-based model of CML, which is well-established in the field of studying the pathogenesis of CML. Additionally, the authors successfully concluded that the MSI2 RNA binding domain RRM1 has a preferential impact on the growth of blast crisis CML.

The quality of this research article could be significantly enhanced by addressing the following points:

Major:

(1) Do mice with BCR-ABL/MSI2-HOXA9 leukemia have an increased pool of leukemic stem cells (LSC), or do they have an increased propensity to develop blast cells? Is it the number of LSCs that has increased, or is it the function of LSC to give rise to the disease that has increased? It is not clear if the detected differences in Lineage-negative cells (Figure S1D) were detected in vitro in retrovirally transduced cells or were detected in vivo in transplanted mice. If the differences were detected in vitro, could the author confirm the same findings in vivo? This will greatly enhance the understanding of in vivo disease pathogenesis and could directly link the aggressivity of the disease (shortened survival) with an increased stem cell-like population.

(2) The authors suggest that BCR-ABL/MSI2-HOXA9 leads to the development of blast crisis-CML. One of the main characteristics of blast crisis-CML is drug resistance. Is BCR-ABL/MSI2-HOXA9 leukemia resistant to classical CML treatment drugs?

(3) The authors have emphasized the heightened expression of Polrmt in delineating the mitochondrial phenotype of BCR-ABL/MSI2-HOXA9 leukemia cells. However, the regulatory mechanism governing the expression of Polrmt by MSI2-HOXA9 has not been clearly demonstrated by the authors. Unveiling this mechanism would constitute a novel finding and significantly elevate the quality of the research.

(4) Did the authors observe any survival differences between BCR-ABL/NUP98-HOXA9 and BCR-ABL/MSI2-HOXA9?

---

## [Referee Report · Reviewer #2 (Public Review)]

The manuscript titled, "Identification of a Musashi2 translocation as a novel oncogene in myeloid leukemia" by Spinler et al. studies the functional role of the translocation t(7;17)(p15;q23), resulting in MSI2/HOXA9 fusion gene, as a secondary driver in bcCML. MSI2-HOXA9 forced expression along with BCR-ABL enhances colony formation and leads to a more aggressive disease in vivo. Depletion of the RNA binding domain RRM1 or RRM2 of MSI2 led to a significant reduction in colony formation, with RRM1 depletion specifically impacting differentiation and blast cell counts. Mechanistically, the authors find that MSI2-HOXA9 aberrantly localizes to the nucleus, elevating the expression of mitochondrial polymerase Polrmt, thereby leading to upregulation of mitochondrial components and enhancing mitochondrial function and basal respiration. Overall, this study examines how the rare MSI2-HOXA9 fusion gene can act as a novel cooperating oncogene and could serve as a secondary hit in the progression of CML to blast crisis.

Strengths:

(1) Demonstration that MSI2-HOXA9 contributes to oncogenesis in the BCR-ABL context.

(2) Development of a novel cooperativity model for BCR-ABL and provides additional supporting data for the role of MSI2 in leukemogenesis.

(3) Evidence that MSI2-HOXA9 acts uniquely compared to MSI2 alone through nuclear vs. cytoplasmic localization and activation of mitochondrial polymerase Polrmt.

Weaknesses:

(1) MSI2-HOXA9 fusion is extremely rare as it has been only found in a handful of patients and it is not clear whether other MSI2 fusions function in a similar manner.

(2) The mechanism needs to be strengthened since MSI2 alone or the HOXA9 mutant may not be linked to the mitochondrial mechanism.

(3) It is not clear that the mitochondrial pathway is sufficient for the MSI2-HOXA9 oncogenic mechanism.

---

## [Author Response]

We are grateful to the reviewers for their interest and enthusiasm about the work, and deeply appreciate their constructive comments and suggestions. Our responses are below

(1) Do mice with BCR-ABL/MSI2-HOXA9 leukemia have an increased pool of leukemic stem cells (LSC), or do they have an increased propensity to develop blast cells? Is it the number of LSCs that has increased, or is it the function of LSC to give rise to the disease that has increased? It is not clear if the detected differences in Lineage-negative cells (Figure 1—figure supplement 1D) were detected in vitro in retrovirally transduced cells or were detected in vivo in transplanted mice. If the differences were detected in vitro, could the author confirm the same findings in vivo? This will greatly enhance the understanding of in vivo disease pathogenesis and could directly link the aggressivity of the disease (shortened survival) with an increased stem cell-like population.

We find that BCR-ABL/MSI2-HOXA9 leads to a marked increase in Lineage negative (Lin-) cells which contains the LSC fraction. Specifically, the LSC containing fraction represented 14.1% of the BCR-ABL driven disease and 56.7% of the BCR-ABL and MSI2-HOXA9 driven disease (p<.0001). This suggests that MSI2-HOXA9 triggers the expansion of the undifferentiated LSC containing pool. In addition, the blast frequency was also increased albeit to a lesser extent, with 63.8% blasts (SEM 1.1) for BCR-ABL and 83.3% (SEM 3.1) for BCR-ABL/MSI2-HOXA9 (p=.0001). This suggests that the resulting aggressive disease seen with MSI2-HOXA9 is a consequence of a large increase in undifferentiated LSC containing cells, as well as the resulting increase in the blast count. The Lin- cells were analyzed from fully established leukemias in vivo (Figure 1—figure supplement 1D)

(2) The authors suggest that BCR-ABL/MSI2-HOXA9 leads to the development of blast crisis-CML. One of the main characteristics of blast crisis-CML is drug resistance. Is BCR-ABL/MSI2-HOXA9 leukemia resistant to classical CML treatment drugs?

The sensitivity to Imatinib is a very interesting question. In general, while differentiated cells in CML are sensitive to Imatinib, the more undifferentiated cells (LSCs) are resistant^1,2^. Based on the fact that therapy resistance in blast crisis is largely driven by the undifferentiated fraction of leukemia cells, and given that BCR-ABL/MSI2-HOXA9 driven disease harbors a larger fraction of these undifferentiated cells, we would predict that BCR-ABL/MSI2-HOXA9 leukemia would also be more resistant to imatinib. However, this would need to be experimentally demonstrated and is an important question to address.

(3) The authors have emphasized the heightened expression of Polrmt in delineating the mitochondrial phenotype of BCR-ABL/MSI2-HOXA9 leukemia cells. However, the regulatory mechanism governing the expression of Polrmt by MSI2-HOXA9 has not been clearly demonstrated by the authors. Unveiling this mechanism would constitute a novel finding and significantly elevate the quality of the research.

Since Polrmt and mitochondrial genes are transcribed in the nucleus we explored whether MSI2-HOXA9 may control mitochondrial gene expression by triggering expression of Polrmt and other key transcription factors. Consistent with this possibility, MSI2-HOXA9 was preferentially found in the nucleus relative to MSI2. In addition, there were 10 occurrences of the minimal MSI2 RRM1 consensus binding sequence UAGU within the Polrmt transcript. While this is consistent with the possibility that Polrmt expression can be post-transcriptionally modulated by MSI2-HOXA9, this needs to be experimentally validated using Clip Seq analysis with wild type MSI2 as well as the MSI2-HOXA9 fusion protein in context of blast crisis CML.

(4) Did the authors observe any survival differences between BCR-ABL/NUP98-HOXA9 and BCR-ABL/MSI2-HOXA9?

In previous work from our lab we have found that the median survival for BCR-ABL/NUP98-HOXA9 was 17 days, and with BCR-ABL/ MSI2-HOXA9 was 18.5 days (p value of 0.22). This suggests that there is not a significant difference in survival times between the leukemias driven by the distinct alleles, and they may be equally aggressive.

(1) MSI2-HOXA9 fusion is extremely rare as it has been only found in a handful of patients and it is not clear whether other MSI2 fusions function in a similar manner.

We were very surprised and excited to see the large number of translocations in solid cancers that involve MSI2. Interestingly, MSI2 translocations occurred both at the N and the C terminus. Distinct translocations are likely to have unique roles in each disease context. For example, if MSI2’s 5 prime end is part of a translocation, it may functionally contribute via its promoter to drive expression in immature cells and could thus activate oncogenic signals (e.g. controlled by the partner gene) in immature cells which are inherently more susceptible to transformation (Eµ-myc is an example of such a translocation). If Msi2’s RRM domains are part of the fusion, they could bind and target RNAs aberrantly (such as in the wrong cell and the wrong time) and lead to activation of downstream oncogenic mediators. To fully understand the role of each of these translocations in each specific cancer, we would need to experimentally test their impact by ectopic expression in the appropriate cell of origin and domain mapping the basis of any impact in the relevant cancer models as we have done for MSI2-HOXA9 in blast crisis CML in the work we report here. While this is an intensive undertaking, it is nonetheless important future work as it will undoubtedly lead to new insight about MSI2 linked translocations in diverse solid cancers such as breast cancer and lung cancer.

(2) The mechanism needs to be strengthened since MSI2 alone or the HOXA9 mutant may not be linked to the mitochondrial mechanism. (3) It is not clear that the mitochondrial pathway is sufficient for the MSI2-HOXA9 oncogenic mechanism.

Our observation that MSI2-HOXA9 triggered changes in mitochondrial function was of particular interest as it was (to our knowledge) uncharted in context of Msi2 signaling in cancer, thus leading us to explore this further. However, multiple other signals are likely downstream regulators and these may well act cooperatively with, or independently of, the heightened­­ mitochondrial function we report here. Among these pathways, the most likely mediators included oncogenic programs related to the Wnt pathway including Wnt, Fzd 3 and Frat1, and those related to the Notch pathway including Tribbles and Hey1 as well as other stem cell genes such as Aldh1. These programs have been previously implicated in the regulation of myeloid leukemia ^3-11^ and could well mediate the impact of the MSI2-HOXA9 translocation. The relative contribution of mitochondrial metabolism and that of developmental and stem cell signals to the onset of MSI2-HOXA9 driven blast crisis CML is an important avenue of future work.

References

(1) Corbin, A. S., Agarwal, A., Loriaux, M., Cortes, J., Deininger, M. W. & Druker, B. J. 2011. Human chronic myeloid leukemia stem cells are insensitive to imatinib despite inhibition of BCR-ABL activity. J Clin Invest 121: 396-409. PMC3007128.

(2) Graham, S. M., Jørgensen, H. G., Allan, E., Pearson, C., Alcorn, M. J., Richmond, L. & Holyoake, T. L. 2002. Primitive, quiescent, Philadelphia-positive stem cells from patients with chronic myeloid leukemia are insensitive to STI571 in vitro. Blood 99: 319-325.

(3) Gurska, L. M., Ames, K. & Gritsman, K. 2019. Signaling Pathways in Leukemic Stem Cells. Adv Exp Med Biol 1143: 1-39. PMC7249489.

(4) Narendra, G., Raju, B., Verma, H. & Silakari, O. 2021. Identification of potential genes associated with ALDH1A1 overexpression and cyclophosphamide resistance in chronic myelogenous leukemia using network analysis. Med Oncol 38: 123.

(5) Ran, D., Schubert, M., Pietsch, L., Taubert, I., Wuchter, P., Eckstein, V., Bruckner, T., Zoeller, M. & Ho, A. D. 2009. Aldehyde dehydrogenase activity among primary leukemia cells is associated with stem cell features and correlates with adverse clinical outcomes. Exp Hematol 37: 1423-1434.

(6) Reya, T., Duncan, A. W., Ailles, L., Domen, J., Scherer, D. C., Willert, K., Hintz, L., Nusse, R. & Weissman, I. L. 2003. A role for Wnt signalling in self-renewal of haematopoietic stem cells. Nature 423: 409-414.

(7) Riether, C., Schürch, C. M., Bührer, E. D., Hinterbrandner, M., Huguenin, A. L., Hoepner, S., Zlobec, I., Pabst, T., Radpour, R. & Ochsenbein, A. F. 2017. CD70/CD27 signaling promotes blast stemness and is a viable therapeutic target in acute myeloid leukemia. J Exp Med 214: 359-380. PMC5294846.

(8) Riether, C., Schürch, C. M., Flury, C., Hinterbrandner, M., Drück, L., Huguenin, A. L., Baerlocher, G. M., Radpour, R. & Ochsenbein, A. F. 2015. Tyrosine kinase inhibitor-induced CD70 expression mediates drug resistance in leukemia stem cells by activating Wnt signaling. Sci Transl Med 7: 298ra119.

(9) Venton, G., Pérez-Alea, M., Baier, C., Fournet, G., Quash, G., Labiad, Y., Martin, G., Sanderson, F., Poullin, P., Suchon, P., Farnault, L., Nguyen, C., Brunet, C., Ceylan, I. & Costello, R. T. 2016. Aldehyde dehydrogenases inhibition eradicates leukemia stem cells while sparing normal progenitors. Blood Cancer J 6: e469. PMC5056970.

(10) Yin, D. D., Fan, F. Y., Hu, X. B., Hou, L. H., Zhang, X. P., Liu, L., Liang, Y. M. & Han, H. 2009. Notch signaling inhibits the growth of the human chronic myeloid leukemia cell line K562. Leuk Res 33: 109-114.

(11) Kang, Y. A., Pietras, E. M. & Passegué, E. 2020. Deregulated Notch and Wnt signaling activates early-stage myeloid regeneration pathways in leukemia. J Exp Med 217. PMC7062512.